# Promoting ordering degree of intermetallic fuel cell catalysts by low-melting-point metal doping

Ru-Yang Shao[1,4], Xiao-Chu Xu[2,4], Zhen-Hua Zhou[2], Wei-Jie Zeng[2], Tian-Wei Song[2], Peng Yin[1], Ang Li[2], Chang-Song Ma[2], Lei Tong[1,2], Yuan Kong [1,3] & Hai-Wei Liang [1,2]

Carbon supported intermetallic compound nanoparticles with high activity and stability are promising cathodic catalysts for oxygen reduction reaction in proton-exchange-membrane fuel cells. However, the synthesis of intermetallic catalysts suffers from large diffusion barrier for atom ordering, resulting in low ordering degree and limited performance. We demonstrate a low-melting-point metal doping strategy for the synthesis of highly ordered $L1_0$-type M-doped PtCo (M = Ga, Pb, Sb, Cu) intermetallic catalysts. We find that the ordering degree of the M-doped PtCo catalysts increases with the decrease of melting point of M. Theoretic studies reveal that the low-melting-point metal doping can decrease the energy barrier for atom diffusion. The prepared highly ordered Ga-doped PtCo catalyst exhibits a large mass activity of $1.07\,A\,mg_{Pt}^{-1}$ at 0.9 V in $H_2$-$O_2$ fuel cells and a rated power density of $1.05\,W\,cm^{-2}$ in $H_2$-air fuel cells, with a Pt loading of $0.075\,mg_{Pt}\,cm^{-2}$.

Proton-exchange membrane fuel cells (PEMFCs) are considered widely as promising renewable energy applications in response to the target of "carbon neutrality" owing to their high energy conversion efficiency and zero-emission[1]. Currently, the commercial PEMFCs stacks rely heavily on carbon-supported platinum (Pt/C) or PtCo alloy catalysts[2,3]. For future large-scale commercialization, a key developmental target of PEMFCs is to reduce the loading of platinum-group-metal to less than 0.10 g/kW[4]. However, as the content of platinum in the catalyst layer decreases, increased voltage loss inevitably arises, leading to poor PEMFCs performance from the efficiency point of view[5]. To this end, advanced Pt-alloy catalysts with smaller activation polarization voltage loss have been explored because of their higher mass-based activity, which can compensate for the voltage loss in the high potential region for the low-Pt cathode[2,5,6]. The improved electrocatalytic activity of Pt-alloys (e.g., PtCo, PtNi) relative to Pt/C was often ascribed to the compressive strain effect that is associated with the smaller Pt-Pt distance in Pt shells of alloy catalysts, which would weaken the binding energy of oxygen-containing intermediate

species ($E_O$) and thus promote the oxygen reduction reaction (ORR) kinetics[7-9].

Compared to random alloys, atomically ordered intermetallic compounds (IMCs), especially the tetragonal distortion of random PtCo with atom ratio of 1:1 ($L1_0$-type), have attracted increasing attention due to their more compressive unit cells and larger formation enthalpy that endow IMCs catalysts with enhanced ORR activity and durability[9-16]. Recently, it has been experimentally verified that the ORR activity was positively correlated with the ordering degree of the alloy catalysts[17-20], but the ordering degree of most reported so-called "intermetallic" catalysts was considerably low, especially for the ones prepared by the industry-relevant impregnation method[10,18,21]. The challenge of preparing highly ordered IMCs catalysts is fundamentally associated with the competition between the thermodynamic driving force and the height of kinetic energy barriers for the disorder-to-order transition[22]. While lowering the annealing temperature far away from the critical temperature of phase transition ($T_c$) will increase the thermodynamic driving force for the nucleation of ordered phase in

[1]Hefei National Research Center for Physical Sciences at the Microscale, University of Science and Technology of China, Hefei, China. [2]Department of Chemistry, University of Science and Technology of China, Hefei, China. [3]Department of Chemical Physics, University of Science and Technology of China, Hefei, China. [4]These authors contributed equally: Ru-Yang Shao, Xiao-Chu Xu. ✉e-mail: kongyuan@ustc.edu.cn; hwliang@ustc.edu.cn

disordered solid solution, the low-temperature annealing is detrimental to the atom diffusion. The trade-off between nucleation and diffusion with annealing temperature inevitably leads to the sluggish rate of disorder-to-order transition (Fig. 1a).

Here, we demonstrate a low-melting-point metal doping strategy to promote the atom diffusion rate, for the preparation of highly ordered L1$_0$-type M-doped PtCo (M-PtCo, M = Ga, Pb, Sb, Cu for doped catalysts; M = Co for undoped PtCo) IMCs catalysts (Fig. 1a). We identify a strong dependence of ordering degree of the M-PtCo catalysts on the melting point of M: the ordering degree of the prepared catalysts increases gradually from ~17% for the undoped PtCo to 65% for Ga-PtCo. Theoretic calculations confirm that the substitution of Co atoms in PtCo with low-melting-point metal atoms can significantly decrease the height of energy barrier for atom diffusion and thus promote the ordering degree. We note that the low-melting-point metals doping has been reported for the low-temperature preparation of intermetallic materials in the field of magnetism[23–26]. The prepared highly ordered Ga-PtCo IMCs catalysts deliver enhanced ORR activity compared to the undoped PtCo catalysts with lower ordering degree. In particular, the Ga-PtCo IMCs catalysts show a high mass activity of 1.07 A mg$_{Pt}^{-1}$ at 0.9 V$_{iR-corrected}$ in H$_2$−O$_2$ fuel cells, along with only 17.8% loss after the accelerated durability tests. The low-melting-point metal doping strategy is also effective for the preparation of Ga-doped PtFe and PtNi IMCs catalysts with improved ordering degree.

## Results

### Structure characterization of M-PtCo (M = Ga, Pb, Sb, Cu)

Carbon-supported M-PtCo (M = Ga, Pb, Sb, Cu) precursors were obtained by the wet-impregnation of corresponding metal salts onto the carbon black Ketjenblack EC-600JD (KJ600) support with a total metal content of 20 wt% and Pt/Co/M atomic ratio of 1:0.8:0.2, which were then subjected to annealing treatment at 1000 °C under 5 vol% H$_2$/Ar. Undoped PtCo with Pt/Co atomic ratio of 1:1 was also prepared for comparison. X-ray diffraction (XRD) patterns showed that the characteristic reflections of the as-prepared M-PtCo matched well with the ordered L1$_0$-PtCo intermetallic phase (Fig. 1b). The average crystallite size is estimated to be around 4 nm for all the M-PtCo catalysts by the Debye-Scherer equation based on the full width at half maximum of XRD patterns (Table S1). High-angle annular dark-field scanning transmission electron microscopy (HAADF-STEM) observations showed that numerous M-PtCo nanoparticles were uniformly dispersed throughout the carbon supports with similar average particle size (Fig. 1c). Statistical analyses of particle size distribution from the low-magnification HAADF-STEM images were consistent well with the XRD results (Figure S1, S2, and Table S1). Elemental mapping with energy-dispersive x-ray spectroscopy (EDS) indicated that Pt and non-Pt elements were distributed homogeneously in individual M-PtCo nanoparticles with an atomic percent approximately equivalent to the theoretical value (Figures S3–S5 and Table S2).

After subtracting the background, the XRD patterns were analyzed further to quantify the ordering degree (Fig. 1b and Figure S6, see the Methods section for the calculation details of the ordering degree)[18]. The ordering degree of the IMCs catalysts increased from 16.36 ± 5.52% for undoped PtCo to 22.39 ± 5.28% for Cu-PtCo, 34.76 ± 8.04% for Sb-PtCo, 47.06 ± 5.78% for Pb-PtCo, and 65.33 ± 2.67% for Ga-PtCo (Table S3). Specifically, the (111) diffraction peak gradually shifted to the position of almost fully ordered L1$_0$ structure with the increase of ordering degree (Figure S7). We further identified that the ordering degree of the M-PtCo catalysts increased with the decrease of the melting point of M (Fig. 1d). Since all the catalysts showed similar crystallite size, the influence of size-dependent thermodynamic drive force of disorder-to-order transition on the ordering degree could be excluded[27–29].

### Mechanism of melting point-dependent ordering degree

To gain insight into the mechanism of the melting point-dependent ordering degree for the M-PtCo catalysts, we performed Density Functional Theory (DFT) calculations to analyze the influence of metal doping on the disorder-to-order transition from the perspectives of both thermodynamics and kinetics. The structural model was constrained by crystal symmetry and a 1/8 doped-PtCo model was used. We assumed that the disorder-to-order transition could be achieved by sliding atoms of adjacent layers (Figure S8). The DFT calculations revealed that there was a clear downward trend in the diffusion potential of M-PtCo, with the energy barrier order of Co (8.36 eV) > Cu (5.98 eV) > Sb (5.76 eV) > Ga (5.55 eV) > Pb (5.41 eV), although the

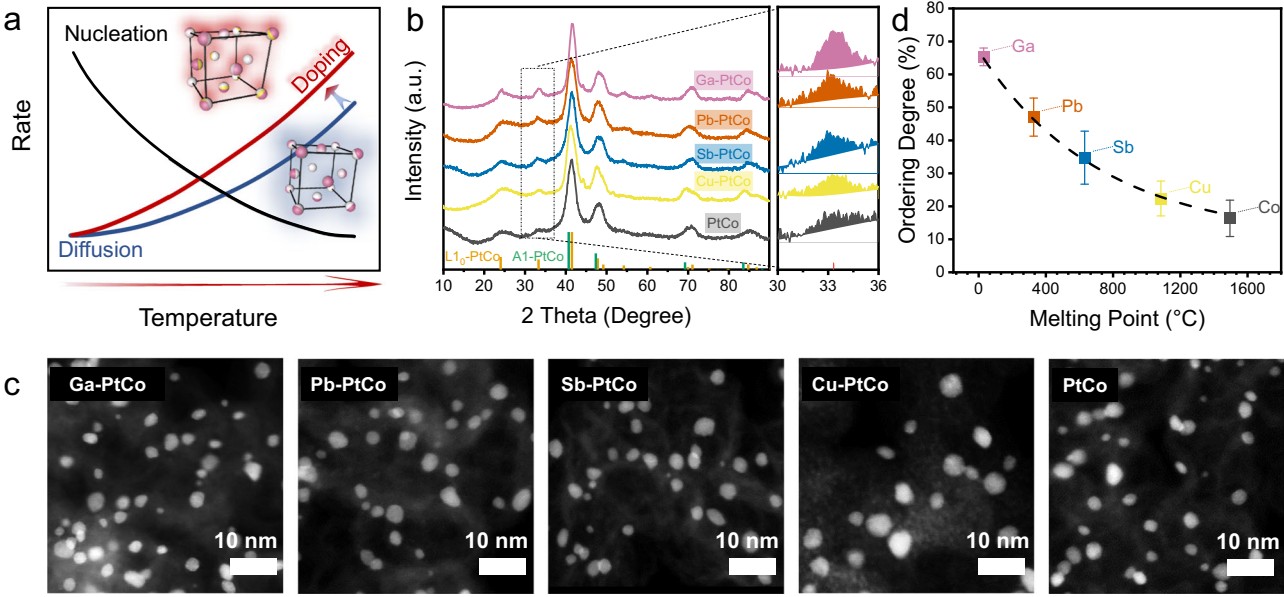

**Fig. 1 | Structural characterization of M-PtCo NPs. a** Schematic illustration showing the dependence of nucleation and diffusion kinetic rate on temperature in the disorder-to-order transition. **b** XRD patterns of M-PtCo (M = Co, Cu, Sb, Pb, Ga), with the highlighted super lattice peak at around 32.8°. The standard peaks for A1 and L1$_0$ PtCo are also shown. **c** HAADF-STEM images of M-PtCo. **d** Ordering degree of the M-PtCo catalysts versus melting point of M. The error bars were obtained by three parallel experiments.

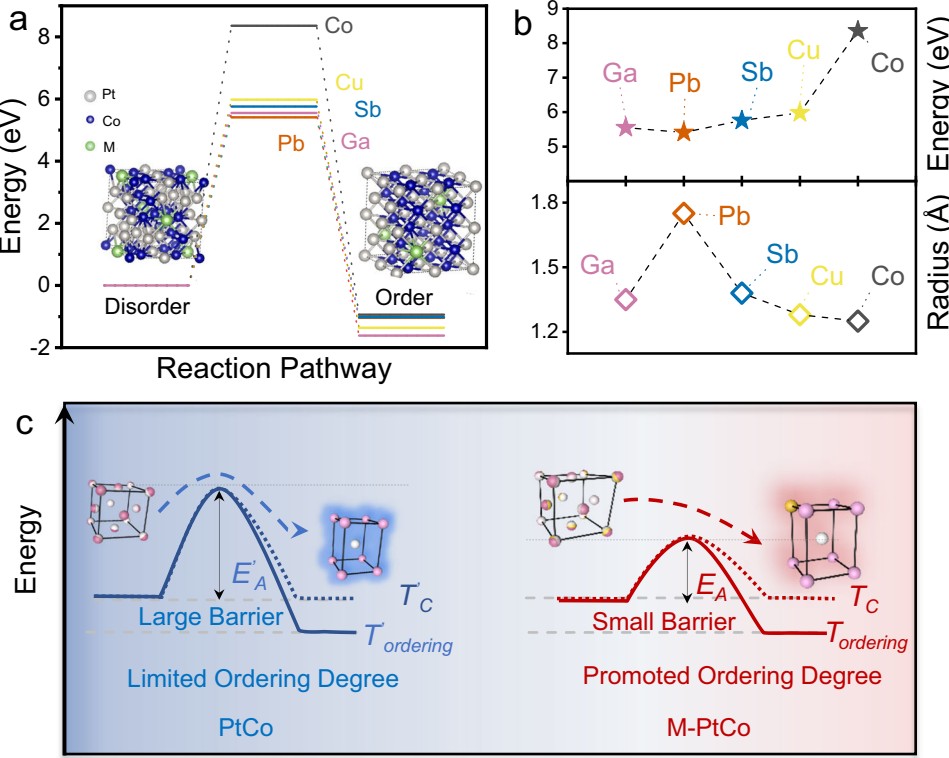

**Fig. 2 | Mechanism of melting-point dependent ordering degree of M-PtCo. a** DFT calculation showing the decreased energy barrier of disorder-order phase transition with the low-melting-point metal doping. The insets are the atomic model for disordered and ordered M-PtCo. **b** Phase transition energy barrier of M-PtCo and atom radius of M. **c** Schematic illustration showing the smaller barrier of disorder-to-order transition for M-PtCo compared to PtCo.

driving force of the phase transition was not much affected by the doped metal (Fig. 2a, b). This result is consistent with the experimental conclusion except for Pb-PtCo. We find that the atomic radii of the other elements are between Co (125 pm) and Pt (139 pm), while the radius of Pb is much larger (175 pm) (Fig. 2b). The larger atomic radius of Pb will lead to increased steric hindrance for the interlayer atomic sliding in the phase transition process[30,31], and therefore suppress the disorder-to-order transition.

On the basis of the above experimental and theoretical studies, we figured out that the promoted ordering degree of the M-PtCo catalysts was associated with the significantly decreased energy barrier of atom diffusion that was induced by the low-melting-metal doping (Fig. 2c). The annealing at a relatively high temperature (i.e., 1000 °C) could ensure the rapid atom diffusion across the support to form well-alloyed but disordered M-PtCo, and the nucleation of ordered $L1_0$-M-PtCo would occur once the sample was cooled to below the critical temperature of phase transition ($T_c$)[20,32]. Further decreasing the temperature far away from $T_c$ could get a larger thermodynamic driving force for the formation of $L1_0$-M-PtCo seeds but slow down the atom diffusion. At a certain temperature ($T_{ordering}$) below $T_c$, the diffusion rate of the disorder-to-order transition was dependent on the difference between diffusion barrier ($E_A$) and the driven force for atom diffusion at $T_{ordering}$. Considering the significantly decreased diffusion barrier upon the low-melting-point metal doping, the diffusion rate in M-PtCo would be larger than that in undoped PtCo, which led to a faster disorder-to-order phase transition in M-PtCo.

## Synthesis of highly ordered IMCs catalysts

To illustrate the universality of the low-melting-point metal doping strategy for the synthesis of highly ordered IMCs catalysts, Ga-doped $L1_0$-PtFe (Ga-PtFe) and $L1_0$-PtNi (Ga-PtNi) were also prepared (Figures S9a and S9b). The ordering degree of Ga-PtFe was 84% and much higher than that of undoped PtFe (26%). The undoped PtNi was

completely disordered structure. The difficulty of obtaining ordered PtNi structure lies in its low $T_c$ (~630 °C), which makes it challenging to balance the trade-off between nucleation and atom diffusion. With the Ga doping, we got the partially ordered Ga-PtNi catalyst with an ordering degree of 30%.

For electrochemical studies, the Ga-PtCo catalyst with a lower Ga/Co atomic ratio of 1:9 (denoted as $Ga_{0.1}$-PtCo) was also prepared in addition to the common one with Ga/Co atomic ratio of 2:8 (denoted as Ga-PtCo). XRD and HAADF-STEM analyses revealed that the prepared $Ga_{0.1}$-PtCo catalyst had similar ordering degree (62%) and average particle size (3.57 nm) to that of Ga-PtCo (Fig. 3a, b). Cs-corrected HAADF-STEM was used to analyze the atomic ordered structure. The alternating arrangement of the brighter dots and darker dots observed along the [1–10] direction represents Pt and Co atom columns, respectively, indicating the $L1_0$-type intermetallic structure (Fig. 3c, e, and Figure S10). Prior to electrochemical tests, the as-prepared Ga-doped PtCo catalysts were treated with 0.1 M $HClO_4$ at 60 °C for 1 h and then annealed at 400 °C for 2 h under $Ar/H_2$ atmosphere to form a highly stable core/shell structures consisting of an intermetallic core and a Pt shell[10,33]. We found that the retention of alloying and ordering degree for the low-Ga-content catalyst (that is, $Ga_{0.1}$-PtCo) was higher than that for the high-Ga-content catalyst (that is, Ga-PtCo) after the acid leaching treatment (Figure S11 and Table S2). The superior structural stability of $Ga_{0.1}$-PtCo over Ga-PtCo was probably owing to the much lower dissolution potential of Ga (−0.549 V) compared to Pt (1.18 V) and Co (−0.28 V)[34,35]. Spherical aberration (Cs)-corrected HAADF-STEM observations revealed that the post-treated $Ga_{0.1}$-PtCo catalyst had a core/shell structure composed of an $L1_0$ PtCo core and a three-atom-layer Pt shell (Fig. 3d, f). EDS mapping showed that the atomic ratio of Pt: Co: Ga of $Ga_{0.1}$-PtCo changed from 1: 0.8: 0.082 to 1: 0.6: 0.019 after acid leaching, which was consistent with the inductively coupled plasma-atomic emission spectrometry (ICP-AES) results (Figure S12 and Table S2). However, severe leaching of Co and Ga was

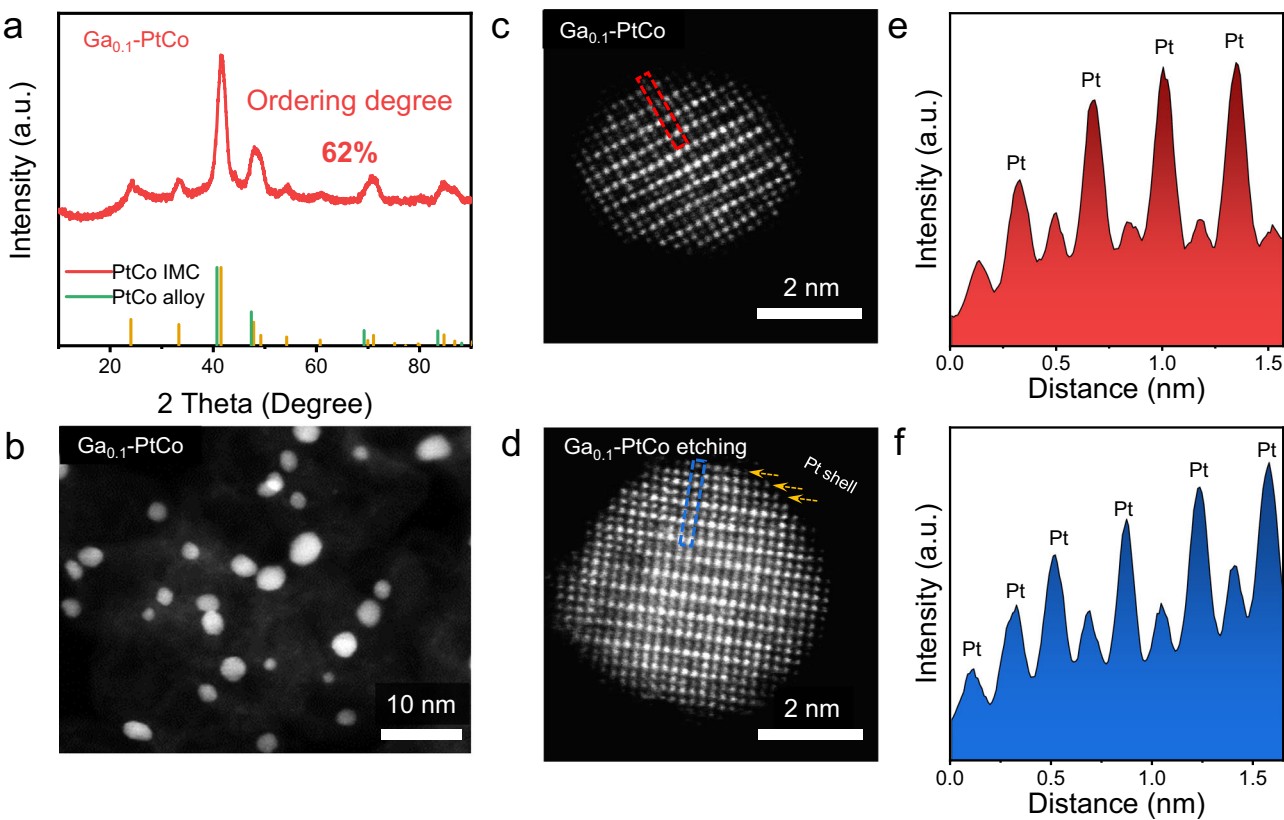

**Fig. 3 | Synthesis and characterization of L1$_0$ Ga$_{0.1}$-PtCo. a** XRD pattern of Ga$_{0.1}$-doped L1$_0$ PtCo. **b** HAADF-STEM image of Ga$_{0.1}$-PtCo NPs. **c, d** Atomic-resolution HAADF-STEM images of untreated Ga$_{0.1}$-PtCo NPs and post-treated Ga$_{0.1}$-PtCo, respectively. **e, f** HAADF line profiles of the marked region in **c, d**.

found for the Ga-PtCo catalyst, in which only 57% Co and 16% Ga were retained after acid leaching.

## Electrochemical performance

We studied the electrocatalytic performance of the treated Ga-PtCo catalysts for ORR by the rotating disk electrode (RDE) and membrane electrode assemblies (MEA) techniques. Undoped PtCo and commercial Pt/C (30 wt%, TKK) were also tested under the same conditions for comparison. Figure 4a displayed the RDE polarization curves of Ga$_{0.1}$-PtCo, Ga-PtCo, PtCo, and commercial Pt/C in O$_2$-saturated 0.1 M HClO$_4$. The mass activity (MA) at 0.9 V$_{RHE}$ of Ga$_{0.1}$-PtCo was calculated to be 2.82 ± 0.23 A mg$_{Pt}^{-1}$, which is higher than 2.30 ± 0.07 A mg$_{Pt}^{-1}$ for Ga-PtCo, 1.45 ± 0.03 A mg$_{Pt}^{-1}$ for PtCo, and 0.40 ± 0.05 A mg$_{Pt}^{-1}$ for commercial Pt/C. The electrochemical surface areas (ECSA) were quantified to be 41.0 ± 1.3 ~ 48.7 ± 2.1 m$^2$ g$_{Pt}^{-1}$ for the alloy catalysts by the CO stripping technique, which were lower than that of Pt/C (69.1 ± 0.67 m$^2$ g$_{Pt}^{-1}$) (Figures S13 and S14). The lower ECSA of the alloy catalysts was associated to their broader particle size distribution and larger volume-weighted average particle size compared to Pt/C (Figure S15 and Table S1). The specific activity (SA) of PtCo, Ga$_{0.1}$-PtCo, Ga-PtCo, and commercial Pt/C were accordingly calculated to be 3.54 ± 0.20, 5.85 ± 0.38, 4.74 ± 0.16, and 0.59 ± 0.08 mA cm$_{Pt}^{-2}$, respectively (Fig. 4b). The higher SA of Ga$_{0.1}$-PtCo over Ga-PtCo and PtCo was contributed to the smaller lattice constant of de-alloyed Ga$_{0.1}$-PtCo (Figure S11), which resulted in a larger compressive strain on the Pt shells and thus promoted ORR kinetics[7,10,14]. The ORR performance of Ga doped or undoped PtFe and PtNi were also studied (Figure S16, Figure S17). The SA of Ga-PtFe, PtFe, Ga-PtNi, and PtNi was tested to be 2.14 ± 0.13, 1.52 ± 0.1, 4.84 ± 0.15, and 3.69 ± 0.38 mA cm$_{Pt}^{-2}$, respectively. We noted that both Ga doped PtFe and PtNi with a higher ordering degree exhibited promoted SA compared to their undoped counterparts. We further calculated the MA

of the catalysts at 0.95 V to avoid the mass transport limited region[36]. The MA at 0.95 V$_{RHE}$ of Ga$_{0.1}$-PtCo was calculated to be 0.345 ± 0.010 A mg$_{Pt}^{-1}$, which is also much higher than that of Ga-PtCo (0.209 ± 0.015 A mg$_{Pt}^{-1}$), PtCo (0.147 ± 0.014 A mg$_{Pt}^{-1}$), and commercial Pt/C (0.071 ± 0.003 A mg$_{Pt}^{-1}$) (Table S4).

We performed DFT calculations to further understand the enhanced ORR activity of the highly ordered Ga$_{0.1}$-PtCo IMCs catalyst. The structure used for calculation was modeled by a slab composed of a fully ordered L1$_0$ Ga-PtCo or A1 PtCo bottom layer that was fixed, and three top Pt layers that were relaxed, denoting as L1$_0$ Ga-PtCo@Pt and A1 PtCo@Pt respectively. Unlike L1$_0$ Ga-PtCo@Pt and Pt, the lattice constant of A1 PtCo@Pt was derived from the high-temperature structure of random PtCo above $T_c$. Thus, the lattice constant of A1 PtCo@Pt was simulated by ab initio molecular dynamics (Figure S18). More details can be found in the Calculation Method section. The DFT calculations revealed that the potential determining step (PDS) of all the three catalysts was the step to form OOH* from OO* with the energy barrier of 0.88, 0.97, and 1.06 eV for L1$_0$ Ga-PtCo@Pt, A1 PtCo@Pt, and Pt, respectively (Fig. 4c). Considering the three-layer Pt shells of the catalysts, the strain effect instead of ligand effect was believed to be the main reason for the changes of electronic structure and ORR activity[8]. We further found a clear correlation between the lattice strain, the d-band center, and the ORR energy barrier: the L1$_0$ Ga-PtCo@Pt showed the lowest d-band center and the largest compressive strain, and thus the highest ORR activity (Fig. 4d, e).

The post-treated Ga$_{0.1}$-PtCo sample was used as the cathode catalyst of MEA to demonstrate the practical application in PEMFCs (Fig. 5a, b). For comparison, the commercial 30 wt% Pt/C (TKK) cathode was also tested under the same conditions. In H$_2$-O$_2$ single-cell test, the MA at 0.9 V of Ga$_{0.1}$-PtCo achieved 1.07 A mg$_{Pt}^{-1}$, which was higher than that of Pt/C (0.41 A mg$_{Pt}^{-1}$) and the US Department of Energy (DOE) 2025 target (0.44 A g$_{Pt}^{-1}$)[4]. The polarization curve was

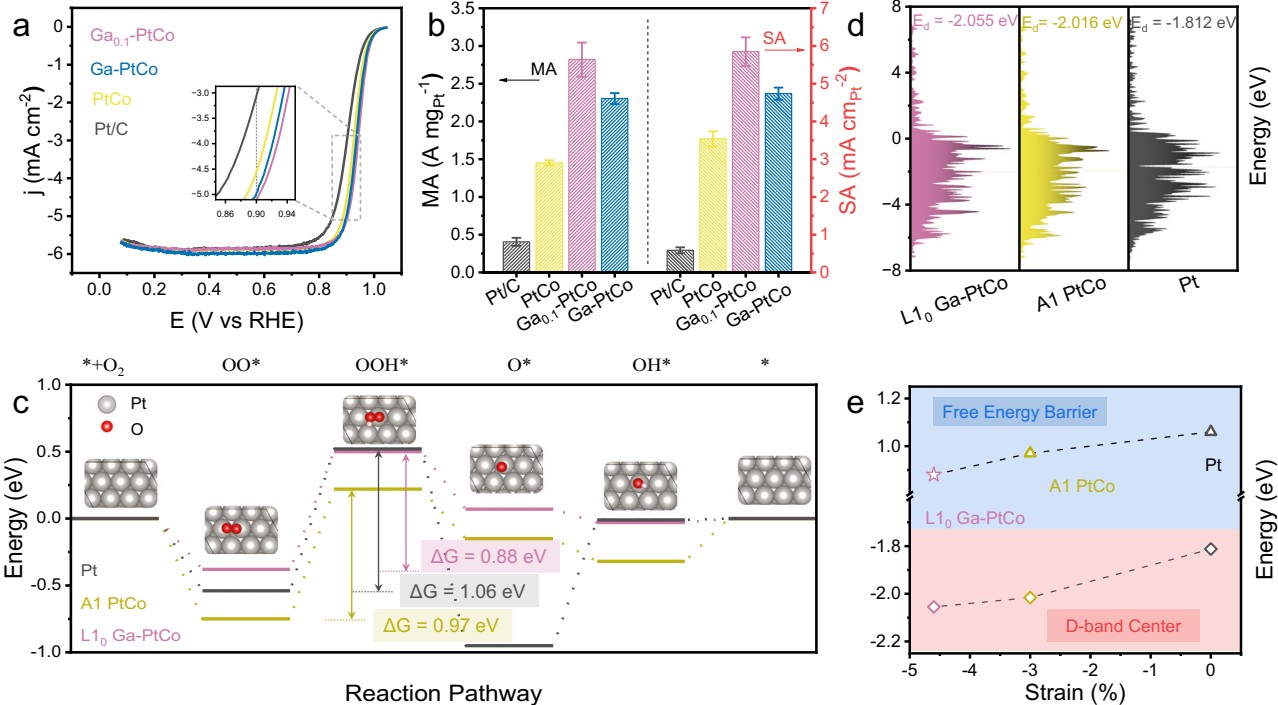

**Fig. 4 | Electrocatalytic performance of Ga-doped PtCo. a** ORR polarization curve of $Ga_{0.1}$-PtCo, Ga-PtCo, PtCo, and commercial Pt/C in $O_2$-saturated 0.1 M $HClO_4$. The insert shows the enlarged polarization curves at around 0.9 V. **b** MA and SA of the catalysts at 0.9 V. The error bar was obtained by three parallel experiments. **c** Theoretical free energy diagram for $L1_0$ Ga-PtCo, A1-PtCo, and Pt based on a three lays Pt shell model from DFT at U = 1.23 V. **d** D-band center of $L1_0$ Ga-PtCo, A1 PtCo, and Pt. **e** Relation between strain, d band center, and ORR free energy barrier of $L1_0$ Ga-PtCo, A1-PtCo, and Pt.

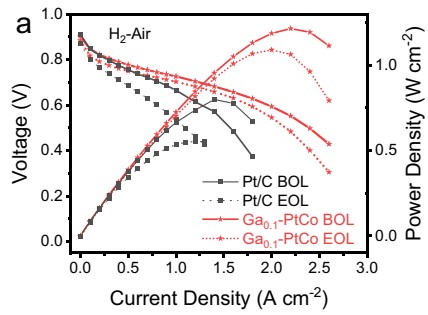

**Fig. 5 | MEA performance of post-treated $Ga_{0.1}$-PtCo. a** $H_2$–air single-cell polarization curves and power density plots of the $Ga_{0.1}$-PtCo and Pt/C cathodes before (BOL) and after (EOL) AST test. Test conditions: cathode loading of 0.075 $mg_{Pt}$ $cm^{-2}$, 80 °C, 100% relative humidity, 150 $kPa_{abs, outlet}$; $H_2$ and air flow rates were fixed at 0.5 and 2.0 liters $min^{-1}$, respectively. **b** MA at 0.9 V in $H_2$-$O_2$ fuel cells and the voltage at 0.8 A $cm^{-2}$ in $H_2$-air fuel cells before and after AST test.

also conducted under $H_2$/Air conditions, in which the peak power density of the MEA made with $Ga_{0.1}$-PtCo cathode was 1200 mW $cm^{-2}$ at 0.55 V with a low Pt loading of 0.075 $mg_{Pt}$ $cm^{-2}$, outperforming the Pt/C cathode (800 mW $cm^{-2}$). The $H_2$-air fuel cell performance of the Ga-PtCo cathode was slightly lower than that of $Ga_{0.1}$-PtCo cathode at high voltage region (Figure S19), owing to the lower ORR activity of the Ga-PtCo catalyst (Fig. 4a, b).

After 30,000 cycles of accelerated stress test (AST) from 0.6 to 0.95 V under $H_2$/$N_2$, the $Ga_{0.1}$-PtCo cathode still maintained a high MA of 0.88 A $mg_{Pt}^{-1}$ with a loss of 17.8%, which meets well the DOE stability target of less than 40% MA loss. The voltage loss of the MEA made with $Ga_{0.1}$-PtCo cathode was 27 mV at 0.8 A $cm^{-2}$ after 30,000 AST cycles, also meeting the DOE 2025 target (<30 mV loss). Further, a chronoamperometric test at a voltage of 0.6 V was conducted to evaluate the long-term durability of the $Ga_{0.1}$-PtCo cathode. The current density of the $Ga_{0.1}$-PtCo cathode dropped by 10% in 100 hours (Figure S20), which could be associated to the accumulation of dissolved Ga or Co cations contaminant in cathode catalyst layer. Continuous

accumulation of dissolved metal cations in the chronoamperometric test could result in increased mass transport resistance and thus performance degradation[37]. Furthermore, the low-Pt $Ga_{0.1}$-PtCo cathode exhibited a high rated power density of 1.05 W $cm^{-2}$ at 0.67 V performed at 94 °C and 250 $kPa_{abs}$ (Figure S21). The high-current-density performance is highly dependent on the transport properties of the cathode catalyst layer, which are controlled by the thickness of the catalyst layer[38], the ionomer distribution[39–41], the Pt roughness factor[2], carbon support type[42], the Pt nanoparticle location[43], as well as the adopted test conditions[39,44] (such as gas flow rate, backpressure, relative humidity, etc.). Systematical optimization of these parameters would significantly promote the transport properties of the catalyst layer and thus improve the $H_2$-air fuel cell performance at high current density in the future.

## Discussion

In summary, we have demonstrated a low-melting-point metal doping strategy for the synthesis of carbon-supported highly ordered

intermetallic M-PtCo catalysts. The ordering degree of the prepared M-PtCo catalysts was inversely correlated with the melting point of the doped M. We understood by experimental and theoretic studies that the doping of low-melting-point metal could greatly decrease the energy barrier of atom diffusion for atom ordering and thus promote the ordering degree. The Ga$_{0.1}$-PtCo catalyst with a high ordering degree of 62% exhibited a large MA and rated power density with a low-Pt loading in PEMFCs. Our results highlight the general validity of low melting point metal doping in the control of phase transition kinetics by decreasing the diffusion barrier, which can further guide the synthesis of diverse intermetallics for fuel cells and other catalysis applications.

## Methods

### Synthesis of the M-PtCo, Ga-PtNi, and Ga-PtFe IMCs catalysts

Carbon-supported L1$_0$-type M-PtCo, Ga-PtNi, and Ga-PtFe IMCs catalysts were prepared by a conventional wet-impregnation method. For M-PtCo, a certain amount of H$_2$PtCl$_6$·(H$_2$O)$_6$, CoCl$_2$·6H$_2$O, and the salt of doped metal were added to the mixture of 100 mg Ketjenblack EC-600J and 60 mL ultrapure water (18.2 MΩ). The total metal content was controlled to be 20 wt%. The mixture was treated with ultrasound for 2 h and stirred for 8 h to get a homogenous solution before drying by rotary evaporation. Finally, the dried precursor was placed in a tube furnace and treated at 1000 °C for 2 h (heating rate, 5 °C/min) in Ar/H$_2$ (95:5) atmosphere to be fully alloyed and then cooling to room temperature with the furnace. The synthesis of L1$_0$ Ga-PtFe and Ga-PtNi were in the same way as M-PtCo.

### Characterization

The XRD patterns were analyzed using a Japan Rigaku DMax-γA rotation anode x-ray diffractometer. The wavelength used in XRD is 1.54178 Å by graphite monochromatized Cu-K radiation. HAADF-STEM images and aberration-corrected HAADF-STEM images were obtained on FEI Talos F200X and JEM ARM200F (S) TEM, respectively, with an accelerating voltage of 200 kV. EDS mapping and line-scanning were carried out using the Super X-EDS system of FEI Talos F200X.

### Ordering degree calculation

The ordering degree was estimated by comparing the normalized intensity of the (110) peak to the sum of the intensities of the (111) and (200) peaks between the experimental XRD patterns and standard Powder Diffraction File cards of L1$_0$-PtCo. The calculation formula is as follows:

$$\text{Ordering degree (\%)} = \frac{S_{(110)}^{exp}}{S_{(111+200)}^{exp}} \bigg/ \frac{I_{(110)}^{PDF}}{I_{(111+200)}^{PDF}} * 100\% \quad (1)$$

$S_{(110)}^{exp}$, $S_{(111+200)}^{exp}$ is the integrated area under (110) peak and the sum of integrated area under (111) and (200) peak of the experimental XRD pattern, respectively. $I_{(110)}^{PDF}$, $I_{(111+200)}^{PDF}$ is the intensity at (110) peak and the sum of the intensity at (111) and (200) peak of the Powder Diffraction File, respectively.

### Calculation methods

The plane-wave technique with a 500 eV cutoff was used for all spin-polarized DFT calculations[45,46], by the Vienna Ab-initio Simulation Package (VASP) codes 5.4[47]. The projected enhanced wave method (PAW)[48,49], and revised Perdew–Burke–Ernzerho (RPBE) exchange-correlation functional[50] were used in the calculations. Spin-polarized calculations were carried out on all surfaces, the electron energy converged to 10$^{-5}$ eV, and the force converged to 0.02 eV/Å. In the single crystal calculation, the transition states (TSs) were located by the climbing image nudged elastic band (CI-NEB) algorithm[51]. In order to maximize the order of the M-PtCo structure, $2 \times 2 \times 2$ PtCo supercell was constructed, in which two Co atoms of sixteen atoms were replaced by M atoms in each supercell. Moreover, M atoms were

symmetrical with respect to the center of the supercell. For AIMD simulation, we performed 5 ps (2500 steps, two fs per step) within the canonical (NVT) ensemble at 1000 K to get an approximate stable structure. The Nose-Hoover thermostat is used in this simulation. Then 5 ps (2500 steps, two fs per step) within the isobaric-isothermic (NPT) ensemble to get reasonable lattice constants at the high temperature. The Langevin thermostat is used in this simulation with the default friction coefficient. (a separate friction coefficient for each of the NTYP atomic species found on the POTCAR-file.) For the ORR simulation, the calculated structure contained two relaxed layers and two layers fixed on the PtCo (111) surface and a $2 \times 2 \times 1$ Monkhorst-Pack k-point mesh was used. The slab model used in our study was constructed based on our experimental results. As shown in Fig. 3d, a 3 atom-layers Pt shell can be observed for the the post-treated Ga$_{0.1}$-PtCo catalyst. Thus, the calculated structure contained three Pt layers epitaxial growth on the PtCo (111) surface. Pt layer adjacent to the Ga$_{0.1}$-PtCo core adopts a lattice parameter closer to that of the core, but outer two Pt shell layers relax towards the lattice constant of bulk Pt, which is known as strain relax[7,52,53]. For the calculation of the Pt (111) system adsorption, two layers of metal are the most common calculation method[54]. L1$_0$ Ga-PtCo(111) is sliced from the ordered Ga-PtCo surface after DFT relaxed structure. A1-PtCo(111) is sliced from the disordered Ga-PtCo surface after DFT relaxed structure with Lattice Parameters from the average of MD process at 1000 K.

All periodic plate calculations were performed with a vacuum separation of at least 13 Å.

### RDE electrochemical measurement

Electrochemical measurements were conducted by using CHI Instruments (CHI 760E) in a three-electrode electrochemical cell. Hg/Hg$_2$SO$_4$ with saturated K$_2$SO$_4$ solution and Pt foil served as the reference electrode and counter electrode, respectively. The reference electrode potential was calibrated to the reversible hydrogen electrode (RHE) potentials under H$_2$-saturated 0.1 M HClO$_4$ solution before tests.

Before the electrochemical test, the as-prepared supported M-PtCo catalysts were etched with 0.1 M HClO$_4$ in oil bath at 60 °C, followed by annealing at 400 °C for 2 h under Ar/H$_2$ to form the stable Pt-skin structure. For the preparation of the working electrode for RDE, 2 mg catalyst was dispersed in isopropanol (1.98 mL) and Nafion (20 μL). Then the mixture was stirred for 2 h and sonicated for 1 h to get a homogeneous ink. After that, a certain amount of the dispersion was dropped onto the rotating disk electrode (RDE, 0.196 cm²) and dried while rotating under ambient temperature. The loading of each catalyst on RDE is 15 μg$_{metal}$ cm$^{-2}$.

All catalyst-coated electrodes were activated by a cyclic voltammetry (CV) method (scan rates of 250 mV s$^{-1}$ and potential ranges of 0.05-1.05 V vs. RHE) until the CV curves completely overlapped. Linear sweep voltammetry (LSV) measurements were then conducted in the O$_2$-saturated 0.1 M HClO$_4$ solution by sweeping the potential from 0.05 to 1.05 V at 10 mV s$^{-1}$ (1600 rpm). For each sample, all tests were repeated three times to get an error bar. For the electrochemical impedance spectroscopy (EIS) measurements, the frequency range was between 0.01 and 100,000 Hz, and initial voltage and amplitude voltage were set at 0.05 and 0.005 V (vs. RHE), respectively. The electrochemical active surface area (ECSA) was obtained by CO stripping test. CO stripping test was conducted by bubbling CO into 0.1 M HClO$_4$ electrolyte for 20 min with holding electrode potential at 0.05 V, followed by bubbling N$_2$ into the electrolyte for 30 min to remove unadsorbed CO in the electrolyte. Then cyclic voltammetry (CV) curve was collected by scanning from 0.05 V to 1.05 V at a scanning rate of 50 mV/s. The ECSA was calculated accordingly:

$$ECSA = \frac{S_{CO}}{0.42 \times v_{scan} \times m_{Pt}} \quad (2)$$

where $S_{CO}$, $v_{scan}$, and $m_{Pt}$ represent the integral area of CO stripping, scan speed, and Pt loading, respectively.

## MEA test

For the preparation of the MEA for PEMFCs, a certain amount of catalyst was dispersed in the mixture of isopropanol and water (1:1). After sonication dispersion, D2020® perfluorosulfonic acid (PFSA) with an equivalent weight of 950 g mol$^{-1}$ was added with the ratio of ionomer to carbon (I/C) of 0.9. The catalyst concentration was controlled to be 2.5 mg mL$^{-1}$. Then the mixture was stirred for 2 h and sonicated for 2 h in the ice bath to get a homogeneous ink. The 5 cm$^2$ catalyst-coated-membrane (CCM) was prepared by ultrasonic spaying the homogenous ink using a SonoTek ultrasonic spray coater at 48 kHz and 90 °C on 12 μm GORE membrane to form 0.075 mg$_{Pt}$ cm$^{-2}$ cathode catalyst layer (0.025 mg$_{Pt}$ cm$^{-2}$ for anode with 30 wt% TKK Pt/C). For the assembly of MEA, 215 μm GDL (22BB, SGL Carbon) and 140 μm gasket were used. The cell endplates were tightened with a torque of 10 Nm to achieve a desired compression of ~22%. Single cells were tested using a Scribner 850e fuel cell test stand. All MEA tests were measured in a 5 cm$^2$ single cell with a graphite flow field containing 7 flow channels in a serpentine arrangement[55]. The pressure drop between the inlet and outlet of the cathode in the flow field under the H$_2$-Air test condition is less than 10 kPa (Table S5).

The MEA was initially activated by scan voltage method from the open circuit to 0.2 V till the polarization curve overlapped under the H$_2$/O$_2$ condition (0.2/0.5 liters min$^{-1}$) at 80 °C, 150 kPa$_{abs}$, and 100% relative humidity (RH). The MA was determined from H$_2$/O$_2$ polarization curve after iR-correction using HFR and H$_2$ crossover correction, and the standardized protocol to evaluate MA is shown in Table S6. The H$_2$-Air polarization curve was conducted under 80 °C, 150 kPa$_{abs}$, and 100% RH, with a gas flow rate of 0.5/2 liters min$^{-1}$ H$_2$/Air for the anode/cathode. Following the guidance of DOE, the accelerated stress test (AST) was carried out at 80 °C, 100 kPa$_{abs}$, and 100% RH with H$_2$/N$_2$ flow of 0.2/0.075 liters min$^{-1}$ for the anode/cathode, including 30,000 cycles of square wave with each cycle holding the MEA at a voltage of 0.6 V for 3 s and then 0.95 V for 3 s. The voltage recovery treatment was conducted to eliminate the effect of contaminants before and after AST test[56]. The rated power density is estimated by H$_2$-Air polarization curve at 94 °C, 250 kPa$_{abs}$, and 100% RH.

## Data availability

Relevant data supporting the key findings of this study are available within the article and the Supplementary Information file. All data presented in this study are available from the corresponding authors (H.-W.L. and Y.K.) upon request. Source data are provided in this paper.

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

## Acknowledgements

We acknowledge the funding support from the National Key Research and Development Program of China (Grant 2018YFA0702001 and 2021YFA1200103), the National Natural Science Foundation of China (Grant 22071225 and 22221003), the Plan for Anhui Major Provincial Science & Technology Project (Grant 202203a0520013 and 2021d05050006), the Fundamental Research Funds for the Central Universities (Grant WK2060190103), the Joint Funds from Hefei National Synchrotron Radiation Laboratory (Grant KY2060000175), Collaborative Innovation Program of Hefei Science Center of CAS (Grant 2021HSC-CIP015), and USTC Research Funds of the Double First-Class Initiative. The numerical calculations in this paper have been done on the supercomputing system in the Supercomputing Center of the University of Science and Technology of China.

## Author contributions

H.-W.L., R.-Y.S., and X.-C.X. conceived and designed the project. R.-Y.S., X.-C.X., Z.-H.Z. synthesized and characterized the catalysts. X.-C.X., W.-J.Z., T.-W.S., P.Y., and L.T. studied the electrochemical performance. A.L. and C.-S.M. made the MEAs. Y.K. conducted the calculation. H.-W. L. and R.-Y.S. co-wrote the manuscript. All the authors discussed the results and commented on the manuscript.

## Competing interests

The authors declare no competing interests.
