## [Peer Review File · Nature Communications]

Promoting Ordering Degree of Intermetallic Fuel Cell Catalysts by Low-Melting-Point Metal DopingREVIEWER COMMENTS

Reviewer #1 (Remarks to the Author):

The paper is very interesting, dealing with the important topic of cathode catalysts for PEMFCs. Although PtCo/C catalysts are now state-of-the-art catalysts, being used also in fuel cell cars, there is room for improvement. This study addresses the doping of PtCo with several metals, demonstrating that the most ordered structure was one characterized by a lower melting point of the doping element (Ga). The amount of Ga was varied (1:9 and 2:8 Ga/Co atomic ratio) and the catalysts were investigated electrochemically in RDE. The most active catalyst (1:9) was tested in single cell and compared with Pt/C showing higher performance and stability. However, in my opinion, since the difference between the 1:9 and 2:8 ratios was not so different in RDE, I suggest testing also the 2:8 catalyst in a fuel cell. It is well known that the results in RDE are not always validated/reproduced under fuel cell conditions.

Furthermore, a short chronoamperometric test (at least 100 h) should be carried out to further validate the stability results obtained by AST.

Minor points:

- please add more details regarding the electrode preparation (type of ionomer, supplier of the GDL, etc.).

-please explain why a very high flow rate was used for the cathode compartment (it should be better to investigate also lower flow rates).

-I suggest (in order to have a complete set of tests) investigating the prepared Ga-PtFe and Ga-PtNi catalysts at least in RDE.

I think that, after these further experiments, the paper could be published in Nat. Comm.

Reviewer #2 (Remarks to the Author):

In this work the authors describe a method for creation of ordered intermetallic catalysts for ORR for PEMFCs. Specifically, they doped PtCo catalysts with Ga, Pb, Sb, and Cu, and claim that this doping reduces the energy barrier for atom diffusion, and thus promote well—ordered catalysts with increased activity. Although some work has been done in the field, it is interesting to see the effect on the performance and durability. While the authors make significant claims regarding the doping effect on the order of the metals in the catalyst, they fail to experimentally convince that they really achieved it. Also, the interpretation of some of the data is somewhat exaggerated. Hence, I cannot recommend publishing it in Nat Comm. Here are some specific comments which may help the authors further improve their work:

- 1) The authors did not calculate the average particle size, but the average crystallite size from the Debye-Scherrer equation. Please make the necessary corrections where needed.
- 2) Higher magnification of the HAADF-STEM is required. In the range of 2-10 nm.
- 3) What is the actual average particle size of the newly made catalysts?
- 4) The peak at 32.8° is very small, and it is not clear how the authors can accurately calculate the ordering degree from there. Other methods are essential to make this case. This is the main claim of this work, and it must be better substantiated!
- 5) Size bar is missing in figure 3D
- 6) How was the atomic percentage measured?
- 7) Why is the ECSA lower for the alloy catalyst? 41 vs. 69 is not a slight difference!
- 8) Mass activity should be calculated at the kinetic region and not the mass transport limited region, there I can assume no difference can be seen.

- 9) The fuel cell performance is low compared to the SOA. What is the reason?
10) Which AST did the authors use? DOE? Please clarify.

Reviewer #3 (Remarks to the Author):

The manuscript reports an original correlation between the degrees of ordering in metal-doped PtCo solids, the melting temperatures of the dopant metals and their catalytic activity for ORR. In the first part of the work, the ordering degrees are proved mainly by XRD and DFT periodic calculations demonstrating that diffusion barriers from disordered to ordered phases correlate inversely with the melting temperature of the doping metals. The finding that doping with low-temperature melting promotes ordering of the bulk phases which in turns improves the catalytic activity is original and of general interest. The overall presentation of the work is sound; however, several points need a better clarification:

- The doping of PtCo alloy resulted in clustering of 2-4 nm nanoparticles of the doped metals in the alloy. How the melting temperature of the bulk metals (the dopants) is related with the melting of the respective small nanoparticles?

- There is insufficient description of how the models for the DFT calculations were built – does the Pt:Co:M (M=dopant) ratio in the models (Fig. S5) respect the experimentally established ratio? What were the scientific arguments to construct the disordered models as depicted in Fig. S5? What could be the diffusion barriers, if other disordered models would have been considered?

- There is a little information on the surface composition, whereas the bulk ordering was thoroughly studied by experimental and theoretical means. In catalysis, the surface should play more important role than the bulk ordering. It seems there is not enough clarification of the surface compositions. Are dopant metals segregating on the surface? Is it possible that the adsorption of oxygen on the surface induces a modification of surface structuring by inducing segregation?

- In the slab model, only two layers were allowed to relax and only PtCo (111) surface orientation was considered. What is the reason to study only this particular surface, if there are not clear experimental evidences presented in the manuscript about the surface compositions and orientations?

- It is not clear why the molecular dynamics, coupled to DFT have been conducted. What are the news results obtained from these dynamics? In Fig. S11, there is only the energy scale, but not a scale for the right part, the cell constant.

- The computational details are not completely described. What is the method used for the ab-initio MD? What approximations for the thermostat and barostat were used? What were the spin-states that led to the minimum energy structures?

Reviewer #1:

The paper is very interesting, dealing with the important topic of cathode catalysts for PEMFCs. Although PtCo/C catalysts are now state-of-the-art catalysts, being used also in fuel cell cars, there is room for improvement. This study addresses the doping of PtCo with several metals, demonstrating that the most ordered structure was one characterized by a lower melting point of the doping element (Ga). The amount of Ga was varied (1:9 and 2:8 Ga/Co atomic ratio) and the catalysts were investigated electrochemically in RDE. The most active catalyst (1:9) was tested in single cell and compared with Pt/C showing higher performance and stability. However, in my opinion, since the difference between the 1:9 and 2:8 ratios was not so different in RDE, I suggest testing also the 2:8 catalyst in a fuel cell. It is well known that the results in RDE are not always validated/reproduced under fuel cell conditions.

Response: Thank you for the reviewer's valuable comment on this issue. Following the reviewer's suggestion, we tested the MEA performance of the Ga-doped PtCo catalyst with Ga/Co ratio of 2:8 (denoted as Ga-PtCo, as shown in Figure R1). We found that the MEA performance of the Ga-PtCo cathode at high voltage region is lower than that of Ga_{0.1}-PtCo cathode. As mentioned in the manuscript, both Ga-PtCo catalysts and Ga_{0.1}-PtCo were treated with 0.1 M HClO₄ at 60 °C for 1 h and annealed at 400 °C for 2 h under Ar/H₂ atmosphere to form highly stable core/shell structures. However, the retention of alloying and ordering degree for Ga-PtCo after the acid leaching treatment was lower than that for Ga_{0.1}-PtCo, which would result in weakened surface compressive strain and thus decreased poor performance for the Ga-PtCo catalyst. Additionally, the poor structure stability of Ga-PtCo is expected to result in severe Ga and/or Co leaching, which will contaminate the ionomer and thus increase the proton transport resistance.

We have added the above data and related discussion in the revised manuscript (Figure S19 and lines 253 ~ 256).

Figure R1. PEMFCs polarization curves and power densities of the Ga_{0.1}-PtCo and Ga-PtCo cathodes.

Furthermore, a short chronoamperometric test (at least 100 h) should be carried out to further validate the stability results obtained by AST.

Response: According to the reviewer's suggestion, we conducted a chronoamperometric test to

validate the stability of the catalysts (Figure R2). The test was conducted at the conditions of 0.6 V, 80 °C, 100% relative humidity, 100 kPa_{abs, outlet}, with the H₂ and air flow rates fixed at 0.2 and 0.5 liters min⁻¹, respectively¹. The current density of the Ga_{0.1}-PtCo cathode dropped by 10% in 100 hours. Continuous accumulation of dissolved Ga or Co cations in the chronoamperometric test would result in enhanced mass transport resistance and continuous performance degradation. In the DOE suggested AST test, we used voltage recovery treatment to eliminate the effect of contaminant before and after AST test².

The above data and discussion have been added in the revised Supplementary Materials (Figure S20 and Lines 261 ~ 267, 388 ~ 389).

Figure R2. Current density as a function of time for Ga_{0.1}-PtCo at 0.6 V in H₂/air. Test conditions: 0.075 mg_{Pt} cm⁻², 80 °C, 100% relative humidity, 100 kPa_{abs, outlet}, with H₂ and air flow rates fixed at 0.2 and 0.5 liters min⁻¹, respectively).

Minor points:

- please add more details regarding the electrode preparation (type of ionomer, supplier of the GDL, etc.).

Response: The ionomer we used for the MEA preparation is D2020® perfluorosulfonic acid (PFSA) with an equivalent weight of 950 g mol⁻¹. The ink was spray-deposited onto 12 μm GORE membrane using a SonoTek ultrasonic spray coater at 48 kHz and 90 °C. The GDL (22BB) with a thickness of 215 μm was provided by SGL Carbon. The thickness of the gasket was 140 μm. To ensure sealing, a torque of 10 Nm was applied to the cell. The GDL compression was calculated to be around 22%. Single cells were tested using a Scribner 850e fuel cell test stand. We have updated above details in the revised manuscript (lines 382 ~393).

-please explain why a very high flow rate was used for the cathode compartment (it should be better to investigate also lower flow rates).

Response: The reason why we used 0.5/2 L min⁻¹ H₂/Air is to avoid mass transport limitations especially at large current density. The discrepancy between the performance observed for electrocatalysts in RDE and in fuel cell is most likely due to the mass transport in the cathode

catalyst layer, such as water transport and O₂ transport. The high flow rate is used to avoid the catalyst layer flooded with liquid water and thus ensure the O₂ transport. According to the reviewer's comment, we further investigated the MEA performance with a lower flow rate (0.2/0.5 L min⁻¹ H₂/Air), as shown in Figure R3. The MEA performance at 0.2/0.5 L min⁻¹ H₂/Air is comparable to that at 0.5/2 L min⁻¹ in the high voltage region (>0.7 V). However, a severe voltage drop occurred with the increase of current density, attributed to limited mass transport³. Optimization of the structures of gas diffusion layer and flow field would promote the mass transport and thus improve the high-current-density performance.

Figure R3. PEMFCs H₂-Air polarization curve and power density of Ga_{0.1}-PtCo at lower flow rates. Test conditions: 80°C, 100% relative humidity, 150 kPa_{abs, outlet}, H₂ and air flow rates were fixed at 0.2 and 0.5 liters min⁻¹, respectively.

-I suggest (in order to have a complete set of tests) investigating the prepared Ga-PtFe and Ga-PtNi catalysts at least in RDE.

Response: Thanks for the reviewer's valuable comment here. As suggested by the reviewer, we further investigated the ORR performance of Ga-PtFe and Ga-PtNi in RDE (Figure R4 and R5). The mass activity (MA) at 0.9 V_{RHE} of Ga-PtFe and Ga-PtNi were calculated to be 1.63 ± 0.16 A mg_{Pt}⁻¹ and 2.29 ± 0.14 A mg_{Pt}⁻¹, which are higher than the counterparts, 1.16 ± 0.02 and 1.75 ± 0.19 A mg_{Pt}⁻¹ for the undoped PtFe and PtNi, respectively. The electrochemical surface areas (ECSA) were quantified to be 75.9 ± 3.2, 76.4 ± 0.8, 47.3 ± 2.3 and 47.4 ± 0.7 m² g_{Pt}⁻¹ for Ga-PtFe, PtFe, Ga-PtNi, and PtNi, respectively. The specific activity (SA) of Ga-PtFe, PtFe, Ga-PtNi, and PtNi were accordingly calculated to be 2.14 ± 0.13, 1.52 ± 0.1, 4.84 ± 0.15, and 3.69 ± 0.38 mA cm_{Pt}⁻². Even though the specific activity of L1₀-PtNi should be higher than L1₀ PtCo considering the surface strain of (111)⁴, the lower ordering degree of Ga-PtNi results in less compressive strain compared to the highly ordered Ga-PtCo catalyst.

We have added the above results and discussion in the revised manuscript (lines 207 ~ 211) and Supplementary Materials (Fig. S16 and S17).

Figure R4. Electrochemical performance of Ga-doped and un-doped PtFe and PtNi. (A) ORR polarization curve of Ga-PtFe and PtFe in O₂-saturated 0.1 M HClO₄. (B, C) CO stripping curves of PtFe and Ga-PtFe, respectively. (D) ORR polarization curve of Ga-PtNi and PtNi in O₂-saturated 0.1 M HClO₄. (E, F) CO stripping curves of PtNi and Ga-PtNi, respectively.

Figure R5. Mass activity, ECSA, and specific activity of Ga-doped and un-doped PtFe and PtNi.

I think that, after these further experiments, the paper could be published in Nat. Comm.

Reviewer #2

In this work the authors describe a method for creation of ordered intermetallic catalysts for ORR for PEMFCs. Specifically, they doped PtCo catalysts with Ga, Pb, Sb, and Cu, and claim that this doping reduces the energy barrier for atom diffusion, and thus promote well—ordered catalysts with increased activity. Although some work has been done in the field, it is interesting to see the effect on the performance and durability. While the authors make significant claims regarding the doping effect on the order of the metals in the catalyst, they fail to experimentally convince that they really achieved it. Also, the interpretation of some of the data is somewhat exaggerated. Hence, I cannot recommend publishing it in Nat Comm.

Response: Thanks for the reviewer's general comments on our work, including the positive comments on the performance and durability of our ordered intermetallic catalysts, as well as the negative comments on the experimental evidence and the interpretation of some data. According to these highly helpful comments, we have performed additional syntheses and characterizations to improve the reliability of our conclusion and interpretation. **Specifically, we have rearranged the XRD data to clearly show that the ratio of signal to noise for the peak at 32.8° was high enough to make the calculation of ordering degree quite reliable. We further repeatedly synthesized and measured the catalyst by three times to further improve the reliability of our data.** Please see our detailed response below.

Here are some specific comments which may help the authors further improve their work:

1) The authors did not calculate the average particle size, but the average crystallite size from the Debye-Scherrer equation. Please make the necessary corrections where needed.

Response: We apologize for misusing the average particle size with the average crystallite size and we thanks to the reviewers for pointing out this mistake. Now we have corrected it in the revised manuscript (lines 84 and 113). In addition to the average crystallite size estimated based on the XRD data, we had also calculated the average particle size from the HAADF-STEM image by statistical analyses of more than 200 particles.

2) Higher magnification of the HAADF-STEM is required. In the range of 2-10 nm.

Response: According to the reviewer's suggestion, the HAADF-STEM images have been replaced with high-magnification ones to better illustrate the average particle size (Figure R6). Additionally, the low-magnification HAADF-STEM images and the statistics of particle size distribution have been moved to Supplementary Materials to illustrate the uniformity of particle distribution (Figure S1 and S2).

Figure R6. HAADF-STEM images of M-PtCo (M = Co, Cu, Sb, Pb, Ga).

3) What is the actual average particle size of the newly made catalysts?

Response: The number-average particle size of Ga-PtCo, Pb-PtCo, Sb-PtCo, Cu-PtCo, PtCo, and Ga_{0.1}-PtCo was estimated to be 3.80 ± 1.32 , 3.43 ± 1.21 , 3.82 ± 1.32 , 4.97 ± 1.37 , 3.58 ± 1.32 , and 3.57 ± 1.34 nm, respectively, based on the statistics from HAADF-STEM images (Figure R7 and Table R1); we further calculated the volume-weighted particle size by the following equation:

$$d_{volume} = \frac{\sum_{i=0}^n d_i * d_i^3}{\sum_{i=0}^n d_i^3}$$

Where n represent the total number of particles used for calculated, d_i represent the particle size of particle i. The calculated values of volume-weighted particle size of the catalysts were listed in Table R1. Moreover, the average crystallite size of Ga-PtCo, Pb-PtCo, Sb-PtCo, Cu-PtCo, PtCo, and Ga_{0.1}-PtCo was also calculated to be 4.07, 3.74, 3.75, 4.21, 4.01, and 4.01 nm, respectively, based on the XRD data (Table R1). These data have been shown in the revised Supplementary Materials (Figure S2 and Table S1). We hope that these updates can make the description of average particle and crystallite size clearer.

Figure R7. Statistics of particle size distribution of M-PtCo (M = Co, Cu, Sb, Pb, Ga) NPs.

Table R1. Structural information of the M-PtCo NPs catalysts.

	XRD Size (nm)	Number-average size (nm)	Volume-weighted size (nm)	Ordering degree (%)	(111) Peak position (°)
Ga _{0.1} -PtCo	4.01	3.57	4.87	62.34	41.74
Ga-PtCo	4.07	3.80	4.98	65.33	41.68
Pb-PtCo	3.74	3.43	4.66	47.05	41.27
Sb-PtCo	3.75	3.82	5.26	34.76	41.49
Cu-PtCo	4.21	4.97	6.01	22.39	41.11
PtCo	4.01	3.58	4.82	16.36	40.87

4) The peak at 32.8° is very small, and it is not clear how the authors can accurately calculate the ordering degree from there. Other methods are essential to make this case. This is the main claim of this work, and it must be better substantiated!

Response: Thanks to the reviewers for pointing out this important issue. The reviewer's comment makes us aware that in the original manuscript it is really difficult to differentiate the intensity of the peak at 32.8° between different samples. This was because we overlapped the XRD patterns of all samples in one figure. We apologize for such cursoriness.

We have rearranged the XRD data in the revised manuscript, particularly, with highlighting the peak at 32.8°. Now it can be clearly found that the intensity of the peak at 32.8° for different sample varies significantly (Figure 1b). More importantly, the ratio of signal to noise for this peak is high enough to make the calculation of ordering degree quite reliable. Compared to disordered PtCo, the superlattice peak (110) around 32.8° is unique for the XRD pattern of L10-PtCo. The extent of ordering can be qualitatively evaluated by the ratio of the integrated area under the (110) peak around 32.8° to that of the sum of the areas under the (111) and (200) peaks around 41.6° and 47.8°. The ratio of fully ordered L10-PtCo is determined by the ratio of the intensity at (110) peak to the sum of the intensity at (111) and (200) peak in the standard L10-PtCo Powder Diffraction File. And the quantitative ordering degree was calculated by comparing the ratio between the experimental XRD patterns and the standard L10-PtCo Powder Diffraction File. The calculation formula is as follows:

$$\text{Ordering degree (\%)} = \frac{S_{(110)}^{exp}}{S_{(111)+(200)}^{exp}} \bigg/ \frac{I_{(110)}^{PDF}}{I_{(111)+(200)}^{PDF}} * 100\%$$

$S_{(110)}^{exp}$, $S_{(111)+(200)}^{exp}$ is the integrated area under (110) peak and the sum of integrated area under (111) and (200) peak of experimental XRD pattern, respectively. $I_{(110)}^{PDF}$, $I_{(111)+(200)}^{PDF}$ is the intensity at (110) peak and the sum of the intensity at (111) and (200) peak of Powder Diffraction File, respectively.

To further improve the reliability of our data, all catalysts were synthesized and measured repeatedly

by three times (Figures R8-R10) and the corresponding ordering degrees were list in Tables R2-R4. The average ordering degree in three measurements for each sample was adopted in the revised manuscript (lines 107 ~ 109, lines 310-318).

Figure R8. XRD patterns of M-PtCo (M = Co, Cu, Sb, Pb, Ga).

Table R2. The ordering degree of M-PtCo (M = Co, Cu, Sb, Pb, Ga) in Figure R8.

	S(110)	S(111)	S(200)	Ordering degree (%)
Ga-PtCo	0.28517	1.64622	0.62199	68.40
Pb-PtCo	0.30529	2.23699	0.8793	53.30
Sb-PtCo	0.24419	2.21893	0.8301	43.57
Cu-PtCo	0.16442	2.37752	0.88691	27.40
PtCo	0.13047	2.37925	1.03675	20.78

Figure R9. XRD patterns of M-PtCo (M = Co, Cu, Sb, Pb, Ga).

Table R3. The ordering degree of M-PtCo (M = Co, Cu, Sb, Pb, Ga) in Figure R9.

	S(110)	S(111)	S(200)	Ordering degree (%)
Ga-PtCo	0.34854	2.09718	0.86287	64.06
Pb-PtCo	0.2554	2.347	0.97008	41.89
Sb-PtCo	0.15093	1.72773	0.76793	32.90
Cu-PtCo	0.11686	1.87728	0.89966	22.89
PtCo	0.12112	2.51325	1.12258	18.12

Figure R10. XRD patterns of M-PtCo (M = Co, Cu, Sb, Pb, Ga).

Table R4. The ordering degree of M-PtCo (M = Co, Cu, Sb, Pb, Ga) in Figure R10.

	S(110)	S(111)	S(200)	Ordering degree (%)
Ga-PtCo	0.30057	1.89256	0.68115	63.54
Pb-PtCo	0.2215	1.81393	0.80715	45.98
Sb-PtCo	0.14094	1.86746	0.88929	27.81
Cu-PtCo	0.10076	2.2944	0.95507	16.87
PtCo	0.05214	2.02759	0.75928	10.18

It is obvious that the ordering degree of M-PtCo increased with the decrease of the melting point of M from above individual XRD pattern. The individual XRD pattern have been updated in the Supplementary Materials (Figure S6). And the integral Figure 1B was also enlarged in the revised manuscript (Figure 1b and Figure R11B).

Figure R11. Structural characterization of M-PtCo NPs.

(A) Schematic illustration showing the dependence of nucleation and diffusion kinetic rate on temperature in the disorder-to-order transition. (B) XRD patterns of M-PtCo (M = Co, Cu, Sb, Pb, Ga). The standard peaks for A1 and L_{10} PtCo are also shown. The attached pattern was the enlarged superlattice peak of M-PtCo. (C) Ordering degree of the M-PtCo catalysts versus melting point of M. (D) HAADF-STEM images of M-PtCo.

5) Size bar is missing in figure 3D

Response: We are sorry for making such a mistake and the size bar in Figure 3d has been added.

6) How was the atomic percentage measured?

Response: Thanks to the reviewers for pointing out this detail. The atomic percentage of the M-PtCo catalysts were measured by both ICP-AES and STEM-EDS (Figure R12 and Figure S5) and the results have been summarized in Table R5 and Table S2.

Figure R12. STEM-EDS spectrum of M-PtCo (M= Co, Cu, Sb, Pb, Ga).

Table R5. Elemental composition of M-PtCo measured by ICP-AES and EDS mapping.

Sample	Metal loading (wt%)	Pt (at%)		Co (at%)		M (at%)	
		EDS	ICP	EDS	ICP	EDS	ICP

Ga-PtCo	21.03	46.99	43.25	41.20	46.65	11.81	10.10
Pb-PtCo	20.76	42.79	46.29	47.62	47.88	9.59	5.83
Sb-PtCo	18.38	48.55	44.97	46.95	48.88	4.50	6.15
Cu-PtCo	17.59	45.34	46.04	43.76	45.18	10.91	8.78
PtCo	19.32	48.48	47.26	51.52	52.74	-	-
Ga _{0.1} -PtCo	20.62	53.12	45.74	42.54	50.10	4.34	4.16

7) Why is the ECSA lower for the alloy catalyst? 41 vs. 69 is not a slight difference!

Response: Thanks for the reviewer's valuable comments here. We agree with the reviewer that the ECSAs between the prepared alloy catalyst and TKK 30wt% Pt/C is obviously different. We have rewritten this part. The lower ECSA of the alloy catalyst compared to TKK 30wt% Pt/C was owing to the larger particle size of the alloy catalysts. The TKK 30wt% Pt/C catalyst is not annealed⁵, while the alloy catalysts presented in this work are prepared at 1000 °C. HAADF-STEM observations confirmed that the average particles size of Ga_{0.1}-PtCo (3.57 nm) was larger than that of TKK Pt/C (2.97 nm) (Figure R13 and R14).

Figure R13. TEM images of TKK 30wt% Pt/C and Ga_{0.1}-PtCo.

Figure R14. Statistics of particle size distribution of TKK 30wt% Pt/C.

Considering that the M-PtCo catalysts were prepared by annealing at a high temperature of 1000 °C, Ostwald ripening and particle migration would happen and result in a broader particle size distribution. Thus, volume-weighted average particle size was calculated to compare the difference between TKK 30wt% Pt/C and M-PtCo. The volume-weighted average particle size of TKK 30wt% Pt/C is 3.36 nm, which is similar to its number average particle size (2.97 nm). Whereas, the volume-weighted average particle size of M-PtCo (4.87 nm) is much larger than its corresponding number average particle size (Table R1). The broader particle size distribution and larger average particle size result in a lower ECSA for M-PtCo compared to Pt/C.

The newly added data and related discussion have been updated to the revised manuscript (lines 199 ~ 201) and Supplementary Materials (Figure S15).

8) Mass activity should be calculated at the kinetic region and not the mass transport limited region, there I can assume no difference can be seen.

Response: Thanks for the reviewer's valuable comment here. The highly ordered M-PtCo with high ORR activity approached the diffusion-limited region at 0.9 V versus RHE, which may lead to an inaccuracy in evaluating the mass activity⁵, just as commented by the reviewer. Accordingly, we additionally calculated the mass activity at 0.95 V versus RHE of catalyst to avoid the mass transport limited region (Figure R15)⁶. The mass activity of Ga_{0.1}-PtCo was calculated to be 0.345 ± 0.010 A mg_{Pt}⁻¹ at 0.95 V, which is higher than 0.209 ± 0.015 A mg_{Pt}⁻¹ for Ga-PtCo, 0.147 ± 0.014 A mg_{Pt}⁻¹ for PtCo, and 0.071 ± 0.003 A mg_{Pt}⁻¹ for TKK Pt/C.

The above data and related discussion have been added to the revised manuscript (lines 211 ~ 215) and Supplementary Materials (Table S4).

Figure R15. Mass activity of Ga_{0.1}-PtCo, Ga-PtCo, PtCo, and TKK Pt/C in O₂-saturated 0.1 M HClO₄ at 0.95V.

9) The fuel cell performance is low compared to the SOA. What is the reason?

Response: Many thanks for the reviewer's valuable comment here. We have compared the MA in H₂-O₂ fuel cells as well as H₂-air fuel cell performance between our Ga_{0.1}-PtCo catalyst with a SOA Pt/C catalyst (TKK, 30 wt% Pt/C) under the exactly identical conditions. In H₂-O₂ single-cell test, the MA of Ga_{0.1}-PtCo at 0.9 V achieved 1.07 A mg_{Pt}⁻¹, which was higher than that of TKK Pt/C (0.41 A mg_{Pt}⁻¹) and the US Department of Energy (DOE) 2025 target (0.44 A g_{Pt}⁻¹). Under the H₂/Air conditions, the peak power density of the MEA made with Ga_{0.1}-PtCo cathode was 1200 mW cm⁻² at 0.55 V with a low Pt loading of 0.075 mg_{Pt} cm⁻², outperforming the TKK Pt/C cathode (800 mW cm⁻²) with the same Pt loading. In addition, the low-Pt Ga_{0.1}-PtCo cathode exhibited a high

rated power density of 1.05 W cm^{-2} at 0.67 V performed at $94 \text{ }^\circ\text{C}$ and $250 \text{ kPa}_{\text{abs}}$ (Figure S21). These results indicate that the prepared highly ordered $\text{Ga}_{0.1}\text{-PtCo}$ catalyst is promising candidate as low-Pt cathode catalyst for practical fuel cell applications.

However, we would not like to directly compare the fuel cell performance of our catalysts with reported values by different groups. Because the fuel cell performance of catalysts, particularly at high current density, is highly dependent on the transport properties of the cathode catalyst layer, which are controlled by many parameters, at least including the thickness of the catalyst layer³, the ionomer distribution (related to I/C ratio⁷, the surface chemistry⁸ and the porous structure of carbon supports⁹), the Pt roughness factor¹⁰, carbon support type (solid versus porous)¹¹, the Pt nanoparticle location (on the carbon surface versus inside)¹², as well as the adopted test conditions (such as gas flow rate¹³, backpressure¹⁴, relative humidity⁷, etc.). Systematical optimization of these parameters would significantly promote the transport properties of the catalyst layer and thus improve the H_2 -air fuel cell performance at high current density, which is however beyond the scope of the present study. In the revised manuscript (Lines 269 ~ 276), we have added our perspective on how to further improve the fuel cell performance by optimizing the transport properties of the catalyst layer in future.

10) Which AST did the authors use? DOE? Please clarify.

Response: Many thanks for the reviewer's valuable comment. We used the AST protocol recommended by the US Department of Energy (DOE), which was carried out at $80 \text{ }^\circ\text{C}$, $100 \text{ kPa}_{\text{abs}}$, and 100% RH with H_2/N_2 flow of $0.2/0.075 \text{ liters min}^{-1}$ for the anode/cathode, including 30,000 cycles of square wave with each cycle holding the MEA at a voltage of 0.6 V for 3 s and then 0.95 V for 3 s.

We have added these details into the revised manuscript (lines 404 ~ 408).

Reviewer #3

The manuscript reports an original correlation between the degrees of ordering in metal-doped PtCo solids, the melting temperatures of the dopant metals and their catalytic activity for ORR. In the first part of the work, the ordering degrees are proved mainly by XRD and DFT periodic calculations demonstrating that diffusion barriers from disordered to ordered phases correlate inversely with the melting temperature of the doping metals. The finding that doping with low-temperature melting promotes ordering of the bulk phases which in turns improves the catalytic activity is original and of general interest. The overall presentation of the work is sound; however, several points need a better clarification:

- The doping of PtCo alloy resulted in clustering of 2-4 nm nanoparticles of the doped metals in the alloy. How the melting temperature of the bulk metals (the dopants) is related with the melting of the respective small nanoparticles?

Response: Thanks for the valuable comment from the reviewer. In this work, we focused on the synthesis of carbon black supported intermetallic PtCo nanoparticle catalysts with promoted ordering degree. We found that the low-melting-point metal doping could effectively promote the ordering degree of PtCo nanoparticles. That is to say, the ordering degree of doped PtCo nanoparticles is higher than that of undoped PtCo nanoparticles prepared under the same conditions. Compared to pure metal M, the melting point of M-PtCo alloy is expected to improve due to the high melting point of Pt¹⁵; but the exact correlation between the type of M and the melting of the M-PtCo nanoparticles is not identified at present. However, combining our experimental findings with DFT calculations, we can understand that the atom diffusion barrier decreased with decreasing melting point of M, which resulted in a significant improvement in the disorder-to-order transition kinetic rate after M doping.

- There is insufficient description of how the models for the DFT calculations were built
- do the Pt:Co:M (M=dopant) ratio in the models (Fig. S5) respect the experimentally established ratio? What were the scientific arguments to construct the disordered models as depicted in Fig. S5? What could be the diffusion barriers, if other disordered models would have been considered?

Response: Thanks for sharing the insightful comment from the reviewer. In our experiment, the atom ratio of Pt:Co:M is 5:4:1. However, in our models, a ratio of 8:7:1 was used, as we had to consider the symmetry of L1₀ M-PtCo. The unit cell of L1₀-PtCo contains 4 atoms and a 2*2*2 supercell should include 16 Co/M atoms. Since the M-PtCo was confirmed to be highly ordered experimentally, we constructed the model with maximum ordering degree, in which the fractional coordinates of M are (X,Y,Z) and (X+0.5,Y+0.5,Z+0.5). In this particular model, the ratio of Co to M is 14:2. While other ratios may be closer, such as 13:3, the symmetry would be completely broken, making it difficult to determine the ground-state.

We have added the sentence below in the Calculation Methods: *'In order to maximize the order of the M-PtCo structure, 2*2*2 PtCo supercell was constructed, in which two Co atoms of sixteen atoms were replaced by M atoms in each supercell. Moreover, M atoms were symmetrical with respect to the center of the supercell.'* (Lines 326 ~ 329).

Figure R16. Atomic movement trajectory of the mixed diffusion model.

We have tried other methods, such as going to only two atomic sites and then changing direction, resulting in four different directions of the mixed diffusion model (see Figure R16). However, we found that this method introduced large bias in the structural changes between the disordered and ordered states. With the current calculation software and transition state algorithm, we were unable to obtain a reasonable convergence result. Unlike the vacancy model that is a binary mixture¹⁶, the possibility of further cavity formation in our ternary mixture is too complex. After trying in our ternary mixture, the results were unreliable.

- There is a little information on the surface composition, whereas the bulk ordering was thoroughly studied by experimental and theoretical means. In catalysis, the surface should play more important role than the bulk ordering. It seems there is not enough clarification of the surface compositions. Are dopant metals segregating on the surface? Is it possible that the adsorption of oxygen on the surface induces a modification of surface structuring by inducing segregation?

Response: Thanks for the reviewer's valuable comment here. Dopant metal could not modify the surface structure of M-PtCo in oxygen reduction reaction, even though the surface structure indeed play a key role in catalysis. This is because the prepared Ga-PtCo catalysts were de-alloyed in 0.1 M HClO₄ and further treated at 400 °C before electrochemical test to form stable PtCo@Pt core-shell structure. That is to say, even for a Pt alloy catalyst, the active surface structure is monometallic Pt. Spherical aberration (Cs)-corrected HAADF-STEM observations confirmed that the post-treated Ga_{0.1}-PtCo catalyst had a core/shell structure composed of an L1₀ PtCo core and a 2~3 atom-layers Pt shell (Figure R17, also the Figure 3d in Manuscript). The compressive strain of surface Pt shell originating from the lattice parameter difference between core and Pt shell determined the ORR performance¹⁷⁻¹⁹. What's more, even Co or Ga segregate into the surface during long time operation (accelerated stability test), they tend to leaching from nanoparticles in the form of Co²⁺ cations rather than on the surface of nanoparticles. Taking the above into account, we therefore did not consider the surface composition specifically in the calculations.

Figure R17. Atomic-resolution HAADF-STEM image and HAADF line profile of the marked region of acid-treated $\text{Ga}_{0.1}\text{-PtCo}$, respectively.

- In the slab model, only two layers were allowed to relax and only PtCo (111) surface orientation was considered. What is the reason to study only this particular surface, if there are not clear experimental evidences presented in the manuscript about the surface compositions and orientations?
 Response: We thank the reviewer for noting this important detail. The slab model used in our study was constructed based on our experimental results. As shown in Figure R17, a 3 atom-layers Pt shell can be observed for the the post-treated $\text{Ga}_{0.1}\text{-PtCo}$ catalyst. Thus, the calculated structure contained three Pt layers epitaxial growth on the PtCo (111) surface. Pt layer adjacent to the $\text{Ga}_{0.1}\text{-PtCo}$ core adopt a lattice parameter closer to that of the core, but outer two Pt shell layers relax towards the lattice constant of bulk Pt, which is known as strain relax²⁰⁻²². For the calculation of the Pt (111) system adsorption, two layers of metal is the most common calculation method¹⁷.

The HAADF-STEM image (Figure R18) results show that the exposed crystal faces of the experimentally prepared $\text{Ga}_{0.1}\text{-PtCo}$ catalyst are (111), (110) and (001) facets. This is reasonable because of the lower specific surface energies of low-index facets. Considering that the nanoparticles are close to the truncated octahedron, the most exposed crystalline surface should be the (111)²³. It is also worth noting that when studying the ORR performance of L1_0 PtCo intermetallic nanoparticles, the mostly studied surface is (111)^{4, 24}.

We have added these details into the revised manuscript (lines 339 ~ 348).

Figure R18. FFT processed atomic-resolution HAADF-STEM images.

- It is not clear why the molecular dynamics, coupled to DFT have been conducted. What are the news results obtained from these dynamics? In Fig. S11, there is only the energy scale, but not a scale for the right part, the cell constant.

Response: We thank the reviewer for sharing the valuable comment. In our study, we employed molecular dynamics simulations to obtain the average lattice parameters at high temperatures and determine the lattice parameter of disordered PtCo, which allowed us to reveal the differences in

strain effects between ordered and disordered PtCo.

We did not calculate the average lattice parameter separately but obtained it indirectly during the simulation. For example, during the first 5 ps, the lattice constant is constant due to the NVT ensemble used. The average lattice constant was determined to be 3.8615 Å during the second 5 ps period after approximately 10 cycles in the NPT ensemble simulation. (Figure R19). We have added the lattice constant in the revised Supplementary Materials (Figure S18).

Figure R19. Lattice constant of disordered PtCo during ab initio molecular dynamics (AIMD) simulation time on the corresponding disordered PtCo.

- The computational details are not completely described. What is the method used for the ab-initio MD? What approximations for the thermostat and barostat were used? What were the spin-states that led to the minimum energy structures?

Response: According to the reviewer's comments here, we have supplemented the computational details in the revised manuscript (lines 331 ~ 336). Specifically, we used the Nose-Hoover thermostat for the first 5 ps NVT ensemble MD simulation, and switched to the Langevin thermostat for the second 5 ps NPT ensemble MD simulation. Spin configuration is primarily determined by software automatically searching for the energy ground state. Consistent with conventional understanding, our calculations also reveal that Pt metal tends towards a spin of zero, while the spin configuration of Co metal tends to favor paramagnetic arrangement. We would also like to clarify that the calculation option of spin polarization is still valid in AIMD (ab initio molecular dynamics).

Reference:

1. Xiao F, *et al.* Atomically dispersed Pt and Fe sites and Pt-Fe nanoparticles for durable proton exchange membrane fuel cells. *Nature Catalysis* **5**, 503-512 (2022).
2. Kabir S, *et al.* Elucidating the Dynamic Nature of Fuel Cell Electrodes as a Function of Conditioning: An ex Situ Material Characterization and in Situ Electrochemical Diagnostic Study. *ACS Appl. Mater. Interfaces* **11**, 45016-45030 (2019).
3. Sassin MB, Garsany Y, Atkinson RW, Hjelm RME, Swider-Lyons KE. Understanding the interplay between cathode catalyst layer porosity and thickness on transport limitations en route to high-performance PEMFCs. *Int. J. Hydrogen Energy* **44**, 16944-16955 (2019).
4. Yang C-L, *et al.* Sulfur-anchoring synthesis of platinum intermetallic nanoparticle catalysts for fuel cells. *Science* **374**, 459-464 (2021).
5. Moriau LJ, *et al.* Resolving the nanoparticles' structure-property relationships at the atomic level: a study of Pt-based electrocatalysts. *iScience* **24**, 102102 (2021).

6. Huang X, *et al.* High-performance transition metal–doped Pt₃Ni octahedra for oxygen reduction reaction. *Science* **348**, 1230-1234 (2015).
7. Liu Y, *et al.* Proton Conduction and Oxygen Reduction Kinetics in PEM Fuel Cell Cathodes: Effects of Ionomer-to-Carbon Ratio and Relative Humidity. *J. Electrochem. Soc.* **156**, B970 (2009).
8. Ott S, *et al.* Ionomer distribution control in porous carbon-supported catalyst layers for high-power and low Pt-loaded proton exchange membrane fuel cells. *Nat. Mater.* **19**, 77-85 (2020).
9. Ramaswamy N, Gu W, Ziegelbauer JM, Kumaraguru S. Carbon Support Microstructure Impact on High Current Density Transport Resistances in PEMFC Cathode. *J. Electrochem. Soc.* **167**, 064515 (2020).
10. Banham D, Ye S. Current Status and Future Development of Catalyst Materials and Catalyst Layers for Proton Exchange Membrane Fuel Cells: An Industrial Perspective. *ACS Energy Lett.* **2**, 629-638 (2017).
11. Liu Y, Ji C, Gu W, Jorne J, Gasteiger HA. Effects of Catalyst Carbon Support on Proton Conduction and Cathode Performance in PEM Fuel Cells. *J. Electrochem. Soc.* **158**, B614 (2011).
12. Ito T, *et al.* Three-Dimensional Spatial Distributions of Pt Catalyst Nanoparticles on Carbon Substrates in Polymer Electrolyte Fuel Cells. *Electrochemistry* **79**, 374-376 (2011).
13. Turhan A, Heller K, Brenizer JS, Mench MM. Quantification of liquid water accumulation and distribution in a polymer electrolyte fuel cell using neutron imaging. *J. Power Sources* **160**, 1195-1203 (2006).
14. Zhang J, Song C, Zhang J, Baker R, Zhang L. Understanding the effects of backpressure on PEM fuel cell reactions and performance. *J. Electroanal. Chem.* **688**, 130-136 (2013).
15. Alloyeau D, *et al.* Ostwald Ripening in Nanoalloys: When Thermodynamics Drives a Size-Dependent Particle Composition. *Phys. Rev. Lett.* **105**, 255901 (2010).
16. Cui M, *et al.* Rapid Atomic Ordering Transformation toward Intermetallic Nanoparticles. *Nano Lett.* **22**, 255-262 (2022).
17. Greeley J, *et al.* Alloys of platinum and early transition metals as oxygen reduction electrocatalysts. *Nat. Chem.* **1**, 552-556 (2009).
18. Hernandez-Fernandez P, *et al.* Mass-selected nanoparticles of Pt_xY as model catalysts for oxygen electroreduction. *Nat. Chem.* **6**, 732-738 (2014).
19. Escudero-Escribano M, *et al.* Tuning the activity of Pt alloy electrocatalysts by means of the lanthanide contraction. *Science* **352**, 73-76 (2016).
20. Strasser P, *et al.* Lattice-strain control of the activity in dealloyed core–shell fuel cell catalysts. *Nat. Chem.* **2**, 454-460 (2010).
21. Wang JX, *et al.* Oxygen Reduction on Well-Defined Core–Shell Nanocatalysts: Particle Size, Facet, and Pt Shell Thickness Effects. *J. Am. Chem. Soc.* **131**, 17298-17302 (2009).
22. Fusy J, Menaucourt J, Alnot M, Huguet C, Ehrhardt JJ. Growth and reactivity of evaporated platinum films on Cu(111): a study by AES, RHEED and adsorption of carbon monoxide and xenon. *Appl. Surf. Sci.* **93**, 211-220 (1996).
23. Atlan C, *et al.* Imaging the strain evolution of a platinum nanoparticle under electrochemical control. *Nat. Mater.*, (2023).
24. Li J, *et al.* Hard-Magnet L10-CoPt Nanoparticles Advance Fuel Cell Catalysis. *Joule* **3**, 124-135 (2019).

REVIEWERS' COMMENTS

Reviewer #1 (Remarks to the Author):

The paper has been revised according to the reviewers' observations. Although some results are still not clear (Figure S20 reports the stability during 100 h, which is quite good, but the current density at 0.6 V is lower in comparison with the polarization curve reported in the text), I suggest accepting the paper for publication.

Reviewer #2 (Remarks to the Author):

Thank you for answering my comments. I am somewhat satisfied with the answers and thus would like to recommend publishing this manuscript in its current form in Nature Comm.

Reviewer #3 (Remarks to the Author):

The authors have addressed the reviewer's concerns and carefully responded to each of them. The manuscript is properly revised accordingly and my recommendation is to be accepted after a minor revision. The sentence "The spin polarization option is still available in AIMD" in line 333 should be deleted, as it does not provide more useful information. Indeed, the AIMD is nothing but a combination between the classical dynamics of the nuclei and the DFT potential energy surface. If spin polarization is implemented for electronic structure calculations and energy gradients, then it is still available in AIMD. My concern was not the spin polarization in the AIMD, but whether the most stable spin state was sought. Apparently, the authors rely on its automatic determination, which in this study is sufficient.

REVIEWERS' COMMENTS

Reviewer #1 (Remarks to the Author):

The paper has been revised according to the reviewers' observations. Although some results are still not clear (Figure S20 reports the stability during 100 h, which is quite good, but the current density at 0.6 V is lower in comparison with the polarization curve reported in the text), I suggest accepting the paper for publication.

Response: Thank you for the reviewer's valuable comment on this issue. The lower current density at 0.6 V in Figure S20 originates from the difference in flow rate and back pressure between the chronoamperometric test and the polarization curve. The H₂-Air polarization curve was conducted under 80 °C, 150 kPa_{abs}, and 100% RH, with a large gas flow rate of 0.5/2 liters min⁻¹ H₂/Air for the anode/cathode. While the gas flow rate for chronoamperometric test was just 0.2/0.5 liters min⁻¹ H₂/Air for the anode/cathode with a back pressure of 100 kPa_{abs} followed the reported test condition (Nature Catalysis 5, 503-512 (2022)). And we have also tested the MEA performance before chronoamperometric test, as shown in Figure R1.

Figure R1. The performance of the MEA used for chronoamperometric test. Test conditions: 0.075 mg_{Pt} cm⁻², 80°C, 100% relative humidity, 150 kPa_{abs, outlet}, H₂ and air flow rates were fixed at 0.5 and 2.0 liters min⁻¹, respectively.

Reviewer #2 (Remarks to the Author):

Thank you for answering my comments. I am somewhat satisfied with the answers and thus would like to recommend publishing this manuscript in its current form in Nature Comm.

Response: Many thanks to the reviewer for recognizing this work!

Reviewer #3 (Remarks to the Author):

The authors have addressed the reviewer's concerns and carefully responded to each of them. The manuscript is properly revised accordingly and my recommendation is to be accepted after a minor revision. The sentence "The spin polarization option is still available in AIMD" in line 333 should be deleted, as it does not provide more useful information. Indeed, the AIMD is nothing but a combination between the classical dynamics of the nuclei and the DFT potential energy surface. If spin polarization is implemented for electronic structure calculations and energy gradients, then it

is still available in AIMD. My concern was not the spin polarization in the AIMD, but whether the most stable spin state was sought. Apparently, the authors rely on its automatic determination, which in this study is sufficient.

Response: Thank you for the reviewer's valuable comment and explanation on this issue. We have deleted the sentence "The option of spin-polarized is still available in AIMD". As the reviewer thought, the most spin state in this study was automatically determined. Once again, we thank the reviewer for recognizing our work.